# Serological and Molecular Survey of Rickettsial Agents in Wild Boars (*Sus scrofa*) from Midwestern Brazil

**DOI:** 10.3390/ani14152224

**Published:** 2024-07-31

**Authors:** Nicolas Jalowitzki de Lima, Gracielle Teles Pádua, Ennya Rafaella Neves Cardoso, Raphaela Bueno Mendes Bittencourt, Mariana Avelar Tavares, Warley Vieira de Freitas Paula, Lucianne Cardoso Neves, Carlos Damian Segovia, Gabriel Cândido dos Santos, Maria Carolina de Azevedo Serpa, Denise Caroline Toledo, Lívia Mendonça Pascoal, Marcelo Bahia Labruna, Alexander Welker Biondo, Felipe da Silva Krawczak

**Affiliations:** 1Veterinary and Animal Science School, Federal University of Goiás, Goiânia 74605-220, Brazil; jalowitzki@discente.ufg.br (N.J.d.L.); warleyvieira@discente.ufg.br (W.V.d.F.P.); luciannecardoso@discente.ufg.br (L.C.N.); liviapascoal@ufg.br (L.M.P.); 2Department of Preventive Veterinary Medicine and Animal Health, School of Veterinary Medicine and Animal Science, University of São Paulo—USP, São Paulo 05508-270, Brazillabruna@usp.br (M.B.L.); 3Department of Veterinary Medicine, Federal University of Paraná—UFPR, Curitiba 81531-970, Brazil; abiondo@ufpr.br

**Keywords:** *Amblyomma*, immunofluorescence assay, *Rickettsia*, *Sus scrofa*, Cerrado biome

## Abstract

**Simple Summary:**

Wild boars (*Sus scrofa* L.) are considered among the most harmful invasive species worldwide, acting as zoonotic spreaders and reservoirs, threatening human and animal health, and having an important economic impact. Accordingly, the present study has assessed the rickettsial exposure, tick infestation of wild boars, and rickettsial DNA presence in ticks from infested animals from the Cerrado biome in midwestern Brazil. Anti-*Rickettsia* spp. antibodies were detected in serum samples of wild boars by immunofluorescence assay. Ticks collected from culled wild boars were identified as *Amblyomma sculptum*, which all tested negative for rickettsial DNA presence. The present study has provided a reliable sampling seroprevalence and indicated high exposure of Eurasian wild boar to some species of *Rickettsia* spp. from the spotted fever group within the Cerrado biome from midwestern Brazil.

**Abstract:**

Wild boars (*Sus scrofa* L.) are considered among the most harmful invasive species worldwide, causing irreversible ecosystem damage, acting as zoonotic spreaders and reservoirs, threatening human and animal health, and having an important economic impact. Accordingly, the present study has assessed the rickettsial exposure, tick infestation of wild boars, and rickettsial DNA presence in ticks from infested animals from the Cerrado biome in midwestern Brazil. Anti-*Rickettsia* spp. antibodies were detected in serum samples of wild boars by immunofluorescence assay. Overall, 106/285 (37.2%) wild boar serum samples from 13 to 18 (72.2%) municipalities showed seroreactivity to at least one of the four *Rickettsia* spp. antigens tested, the largest number of wild boars serologically tested to *Rickettsia* spp. in this type of study. Among the 106 seroreactive animals, 34 showed possible homologous reactions between *R. parkeri*, *R. amblyommatis,* and *R. bellii*, with endpoint titers between 128 and 512. A sample of 45 ticks collected from four culled wild boars was identified as *Amblyomma sculptum*, and all tested negative for rickettsial DNA presence. In conclusion, this study has provided a reliable sampling seroprevalence and indicated high exposure of wild boars to rickettsial agents, with a potential interaction with *Rickettsia* spp. from the spotted fever group within the Cerrado biome from midwestern Brazil.

## 1. Introduction

The invasion of wild boars (*Sus scrofa* L.) in Brazil extends far beyond the immediate ecological damage. Their presence poses a threat to agricultural productivity, as they raid crops and compete with livestock for food resources. Additionally, wild boars can carry diseases that can be transmitted to both domestic animals and humans, raising concerns for public health. Moreover, the disruption caused by wild boars has significant economic repercussions, affecting tourism and the livelihoods of local communities that depend on ecosystem services [1,2].

Wild boars have been considered an exotic invasive species in Brazil, historically dispersed and reported in all six Brazilian biomes (Amazon, Caatinga, Cerrado, Atlantic Forest, Pampa, and Pantanal), causing a series of harmful impacts on human, animal, and environmental health [2,3,4,5]. This adaptation to a broad spectrum of habitats, characterized by varying vegetation structures, resource availability, and climatic conditions, is absent in their native Eurasian range [3,6]. This highlights a concern for potential ecosystem disruption in Brazil: the establishment of wild boar populations may lead to increased competition with native ungulates (*Dicotyles tajacu* and *Tayassu pecari*) for food resources, predation pressure on smaller fauna, and habitat disturbances through omnivorous foraging behavior [7,8,9].

Wild boars have emerged as significant hosts and reservoirs for various pathogens affecting both animals and humans worldwide, like tuberculosis, influenza virus, *Mycobacterium* spp., *Toxoplasma* spp., and *Trichinella* spp. [10]. Their wide distribution, foraging behavior, and frequent contact with urban environments are what make them potential pathogen spreaders [9,10,11]. The expanding wild boar population, coupled with the intensifying human–animal interface, elevates the risk of zoonotic diseases, the need for stringent population control measures, and epidemiological surveillance [2,12,13,14].

In relation to rickettsial diseases, wild boars play an important role in the spreading and maintenance of *Dermacentor marginatus* tick populations, vector of *Rickettsia slovaca*, from the spotted fever group (SFG) and the causative agent of the tick-borne lymphadenopathy disease [15,16]. Emerging evidence suggested that free-ranging wild boars may also play a significant role in the enzootic cycle of other rickettsial agents, including *Rickettsia monacensis*, *Rickettsia helvetica* in Romania, and probably other neighboring European countries [17].

In Brazil, where at least two SFG agents have been associated with human spotted fever illness, *Rickettsia rickettsii* has presented an average 32% fatality rate nationwide, increasing 50–80% in the endemic southeastern Brazilian region [18,19]. In such areas, disease may cause fever, petechial hemorrhage, and, without prompt and adequate treatment, eventual death. On the other hand, infection by *Rickettsia parkeri* mostly presents a milder infection, with a characteristic inoculation eschar at the tick bite location and no reported deaths to date [18,19].

In the natural environment, the capybara (*Hydrochoerus hydrochaeris*) is regarded as the primary amplifying host for the spotted fever agent, *R. rickettsii* [20]. Conversely, the rice rat (*Euryoryzomys russatus*) has been identified as a competent amplifying host for *R. parkeri*, particularly for immature stages of the tick vector *Amblyomma ovale* [21]. Although there are still no studies that elucidate whether the wild boar is an amplifying host for any rickettsial agent in Brazil, there have been only two studies that evaluated the serological exposure of wild boars to SFG agents [22,23]. These two studies, in conjunction with a third study [24], reported *Amblyomma sculptum* as the most common tick species infesting wild boars in the Cerrado biome. Because *A. sculptum* is the main vector of *R. rickettsii* to humans in Brazil, wild boars might be suitable sentinels for spotted fever illness [23]; also, *A. sculptum* is widely distributed throughout the Goiás State [25].

Accordingly, the present study aimed to determine the seroprevalence of *Rickettsia* spp. in a large sample of wild boars, the largest sample to date for this type of study in South America. In addition, some ticks collected from wild boars were tested for the presence of *Rickettsia* spp.

## 2. Materials and Methods

### 2.1. Study Area

The study herein obtained data from 18 to 246 (7.31%) municipalities of Goiás State (Figure 1) located in the Cerrado biome, midwestern Brazil. The Cerrado biome is a tropical region with phytoecological characteristics predominantly similar to the savannah but with areas of seasonal deciduous forest, seasonal semideciduous forest, and savannah–forest ecotones. The Cerrado region has two distinct seasons: a summer rainy season (October to April) and a winter dry season (May to September) [26].

In Brazil, Goiás State has been ranked 11th in population with 7 million inhabitants, 9th in Gross Domestic Product (GDP) with a value of USD 5.3 billion, and 9th in Human Development Index (HDI) with a score of 0.737 (high) in 2023 [27].

### 2.2. Sampling

Hunting wild boars has been permitted in Brazil due to their status as an invasive exotic fauna, with the coordination of this activity falling under the responsibility of the Agricultural Defense Agency (Agrodefesa) in the Goiás State. Overall, 285 blood samples were collected from culled free-ranging wild boars in different locations within Goiás State in collaboration with exotic fauna controllers (i.e., hunters) between 2018 and 2023. Following the placement of tubes in appropriate packaging, the samples were transported to the local Agricultural Defense Agency, which immediately shipped the samples to the University Laboratory. Serum samples were aliquoted and stored at −20 °C until they could be processed for serological analysis. In addition, a convenience sample of 45 ticks was collected from four free-ranging wild boars that were culled in the municipalities of Silvânia, Edealina, and São Miguel do Passa Quatro. This was conducted over the course of eight on-field incursions, which lasted approximately 67 h. The rationale behind the decision to sample only four animals for tick collection was based on the inherent unpredictability and limited productivity of on-field sampling of free-ranging wild boar.

### 2.3. Detection of Antibodies against Rickettsia spp.

Wild boar serum samples were individually tested by immunofluorescence assay (IFA) using crude antigens fixed on slides, derived from the following four isolates of *Rickettsia* spp. from Brazil: *R. rickettsii* strain Pampulha [28], *R. parkeri* strain At24 [29], *Rickettsia bellii* strain Mogi [30], and *Rickettsia amblyommatis* strain Ac37 [31]. Fluorescein isothiocyanate-labeled rabbit anti-pig IgG was used as conjugate at a dilution of 1:400 (whole molecule, IgG, Sigma Diagnostics), as described in the literature [23]. Initially, the sera were individually screened at a 1:64 dilution in phosphate-buffer saline (PBS) against each of the previously described *Rickettsia* antigens. In case of a positive reaction in the screening dilution, the samples were tested at two-fold serial dilutions to determine the endpoint titers for each of the four *Rickettsia* antigens. If one serum sample showed an endpoint titer at least four-fold higher for one *Rickettsia* antigen than the endpoint titers elicited to the other three *Rickettsia* antigens, it was considered possibly homologous to the former rickettsial agent or to a very closely related genotype [32,33,34]. On each slide, a non-reactive serum (negative control) and reactive serum (positive control) from wild boars from another study [23] were tested at a 1:64 dilution.

Data were calculated as prevalence (%) with their respective confidence intervals (95% CI).

### 2.4. Ticks and Molecular Detection of Rickettsia

The ticks were identified to the species level under a stereomicroscope using descriptions and taxonomic criteria [35] and subjected to DNA extraction by the guanidine-phenol isothiocyanate technique [36]. DNA samples were tested by TaqMan real-time qPCR assay, targeting a 147 bp fragment of the rickettsial citrate synthase (*gltA*) gene [31,37]. To validate the tick DNA extraction protocol, a conventional PCR (cPCR) was performed, targeting the tick mitochondrial *16S rRNA* gene [38]. If a DNA sample did not yield products of the expected size in this cPCR assay, it was discarded from the study.

## 3. Results

### 3.1. Antibodies Anti-Rickettsia spp.

Overall, 37.2% (106/285—31.6%–42.8% CI) of the wild boars were seroreactive (titer ≥ 64) to at least one *Rickettsia* species (Table 1). Only 5 of the 18 municipalities had all wild boars seronegative for *Rickettsia* spp.; however, only one or two animals were tested in each of these five municipalities. In the remaining municipalities, 9.1–100% of the wild boars were reactive to at least one *Rickettsia* species (Figure 1). A total of 27/285 (9.5%), 76/285 (26.7%), 27/285 (9.5%), and 36/285 (12.6%) wild boars were reactive to *R. rickettsii*, *R. parkeri*, *R. bellii*, and *R. amblyommatis*, respectively (Table 1). A total of 17/106 (16.0%) wild boar sera from six municipalities showed titers to *R. parkeri* at least 4-fold higher than those to any of the other three antigens. The antibody titers in these 17 wild boars were considered to have been stimulated by *R. parkeri,* or a very closely related species (Table 1). Based on this criteria, nine sera from five municipalities were considered to have been stimulated by *R. bellii*, and eight sera from two municipalities were considered to have been stimulated by *R. amblyommatis*. For the remaining 72 seroreactive wild boars, it was not possible to determine the possible rickettsial agent involved in a homologous reaction because they displayed similar endpoint titers (<four-fold difference) for two or more *Rickettsia* species or had a single endpoint titer of 64 for a single *Rickettsia* species. Overall, IFA endpoint titers varied from 64 to 256 for *R. rickettsii*, *R. parkeri*, and *R. bellii* and from 64 to 512 for *R. amblyommatis* (Table 1).

### 3.2. Ticks

All ticks were identified as *A. sculptum* (15 females, 30 males). DNA samples were extracted from 33 ticks subjected to qPCR, and all tested negative for the rickettsial *gltA* gene. Twelve ticks were deposited in the tick collection at “Coleção Nacional de Carrapatos do Cerrado” (CNCC) of the Veterinary and Animal Science School, Federal University of Goiás, under accession numbers CNCC81, CNCC82, CNCC83.

## 4. Discussion

Serological analysis of wild boars from 18 municipalities of Goiás State indicated that these animals had been exposed to rickettsial agents within the Cerrado biome, where 37.2% (106/285—31.6%–42.8% CI) of the tested animals were seroreactive to *Rickettsia* spp., with endpoint titers ranging from 64 to 512. Only two studies have evaluated wild boars exposure to *Rickettsia* spp. in Brazil, including a serosurvey of 36 wild boars from Goiás (Cerrado biome) and 44 from Paraná (Atlantic Forest biome) states, resulting in 58/80 (72.5%) seroreactive samples with 64 to 1024 titers [23], and a survey in Minas Gerais State (Cerrado biome) found that 24/31 (77.4%) wild boars had seropositive responses with titers ranging from 64 to 1024 [22]. It is important to emphasize that the highest specific seroreactivity in both wild boars’ studies was for *R. rickettsii*, the agent of Brazilian spotted fever (BSF), while the highest specific seroreactivity in the study herein was for *R. parkeri*, another known pathogenic agent of human spotted fever in Brazil.

The previous studies also used the same four rickettsial antigens used in the present study (*R. rickettsii*, *R. parkeri*, *R. amblyommatis*, *R. bellii*), plus a fifth antigen, *Rickettsia rhipicephali*, all tested by the same IFA protocol used in the present study. Considering each *Rickettsia* species separately, the highest seroreactivity of the present study was for *R. parkeri* (26.7%), contrasting to the highest specific seroreactivity to *R. rickettsii* in the previous two studies: 60% (*R. rickettsii*) in the study of Sousa et al. [22], and 48.7% (*R. rickettsii*) in the study of Kmetiuk et al. [23]. Interestingly, considering only the 36 wild boars from Cerrado in the latter study, the highest seroreactivity was for *R. parkeri* (35.1%), similar to the present study. These results, coupled with the current findings of 17 animals with seroreactivity that may be homologous to *R. parkeri*, suggest that wild boars have been exposed to *R. parkeri* in Goiás State. Additionally, in Minas Gerais, another State within the Cerrado biome, and in the Atlantic Forest biome, there is a heightened exposure to *R. rickettsii* among wild boars [22,23].

Discrepancies in seropositivity rates for *Rickettsia* spp. between the present study and the previous findings [22,23] can be attributed to several factors, including differences in sample size, wild boar populations, and geographic and biome locations. Such differences may significantly impact regional variations of tick species and amplifier hosts, leading to disparate exposure to rickettsial agents. Edaphoclimatic conditions may also influence the presence of tick hosts and, consequently, the rickettsial cycle [18,39]. Therefore, future studies should address the *R. parkeri* ecology and ecoepidemiology, assessing the dynamics between wild boars and rickettsial agents.

Both the present study and the study of Kmetiuk et al. [23] sampled wild boars in the southern region of Goiás State. In this same region, *R. parkeri* has been reported infecting the ticks *Amblyomma ovale* [40,41] and *Amblyomma triste* [42]. Moreover, at least two of these reports of *R. parkeri*-infected ticks were from areas near the Jataí municipality [41,42], where the highest number of seroreactive wild boars were found, possibly homologous to *R. parkeri* (Table 1). Both *A. ovale* and *A. triste* are competent vectors of *R. parkeri* [43,44] and are naturally associated with native wild large mammals, such as tapirs (*Tapirus terrestris*), peccaries (*Tayassu pecari*, *Dicotyles tajacu*), and several Carnivora species [45,46], which are all present in southern Goiás [47]. Because there are previous studies reporting *A. ovale* and *A. triste* infesting free-ranging domestic pigs (*S. scrofa*) and wild boars [24,46,48], it is reasonable to hypothesize that wild boars have been exposed to these two tick species in southern Goiás, which could have resulted in their seroconversion to *R. parkeri*. Indeed, this statement requires further investigation into the dynamic between wild boars and *R. parkeri*, especially because *R. parkeri* is an emerging human pathogen in Brazil [18,19], where wild boars could participate in the ecoepidemiology of the disease in southern Goiás.

Besides *R. parkeri*, nine and eight wild boars elicited seroreactiveness, possibly homologous to *R. bellii* and *R. amblyommatis*, respectively. Although these two agents have not been confirmed as human pathogens [39], they have been the most frequently reported *Rickettsia* species infecting ticks in Brazil, including various *Amblyomma* species from the Cerrado biome, such as *A. sculptum*, *Amblyomma dubitatum*, *Amblyomma rotundatum* and *Amblyomma cajennense* [23,49,50,51,52]. Indeed, *A. sculptum* has been the dominant tick species reported on wild boars in the Cerrado biome [22,23]. Because both *R. amblyommatis* and *R. bellii* have been reported in *A. sculptum* ticks from the Brazilian Cerrado [24,41,53,54], the seroreactiveness of wild boars to these two rickettsial agents could be related to *A. sculptum*. Other *Amblyomma* species cannot be excluded since many of the tick species reported from *S. scrofa* [46] have also been reported infected by *R. amblyommatis* and *R. bellii* [50]. Other rickettsial agents could also be related to the general seroreactiveness of wild boars since most of them demonstrated low endpoint titers or serological cross-reactions with similar endpoint titers. Several other *Rickettsia* species or novel genotypes have been reported in *Amblyomma* ticks (including *A. sculptum*) from the Cerrado biome of Goiás [40,42,55].

Serological results presented here for *R. amblyommatis* may be explained by its common detection in ticks. This rickettsial agent has already been described in 34 tick species worldwide [56], including *A. sculptum*, *Amblyomma oblongoguttatum*, and *Amblyomma scalpturatum*, which have been identified as parasitizing pigs in Brazil [41,57].

Brazilian spotted fever (BSF), the most severe spotted fever illness in the world [39], is caused by *R. rickettsii* and transmitted to humans mainly by *A. sculptum* [58]. The identification of all ticks as *A. sculptum* was expected, given since this tick has been detected in wild boars in Brazil [2,22,24] and the regions from which the specimens were collected (Cerrado biome) [25,59]. Notwithstanding the absence of *Rickettsia* DNA detection in the collected ticks, the potential risk of the circulation of *R. rickettsii* in the area cannot be ruled out, particularly in light of the limited sample size and the observation that less than 1% of *A. sculptum* ticks in nature were infected by *R. rickettsii* [37]. *A. sculptum,* as the dominant tick species detected infesting wild boars in the Cerrado biome by other studies [23,24], points out this mammal species as suitable sentinels for BSF in this biome, as recently proposed [23].

No association could be established between the serological findings pertaining to *R. parkeri* and the tick species identified. However, ticks that act as vectors of this agent have already been documented in wild boars and domestic pigs (*S. scrofa*). *Amblyomma triste* has been described as parasitizing wild boars in Argentina, suggesting that these animals may contribute to the maintenance of this tick species in that environment [48]. *A. ovale* has been documented parasitizing wild boars in transition areas between Cerrado and Atlantic Forest biomes, where hemopathogens of the *Anaplasmataceae* family have been detected [24]. Furthermore, parasitism of *A. ovale* has been reported in domestic pigs from areas of the Amazon Rainforest [57].

Considering the possibility of cross-reaction between rickettsiae in serological tests [39], there is the possibility that our results are involved in the detection of an agent phylogenetically close to *Rickettsia tillamookensis* of the transitional group of rickettsiae associated with *A. sculptum*, recently documented in a conservation area in the southern region of the Goiás State. Serum samples were not tested by IFA for *R. tillamookensis* since this microorganism has not yet been isolated; it has only recently been described for the first time in Brazil [42]. Another possibility is that our serological findings are related with *Rickettsia* sp. strain NOD, identified in *Amblyomma nodosum* in Goiás State, phylogenetically close to *R. parkeri* s.s. [55], which, to date, no suiformes have been documented infected with this microorganism.

Seropositivity for *R. bellii* may be associated with the prior detection of the pathogen in *A. dubitatum*, in addition to the high serological titers in dogs and capybaras within the Goiás State [51]. *A. dubitatum* has been observed parasitizing wild and domestic pigs [48], indicating the possibility of exposure to *R. bellii*.

Our findings demonstrate that only 9.5% (27/285) of the wild boars were seroreactive to *R. rickettsii*, with endpoint titers ranging from 64 to 256. This evidence suggests that the southern region of Goiás State may not be endemic to BSF. This statement is supported by official data from the Brazilian government, which showed that all cases of spotted fever illness in Goiás State (the *Rickettsia* species has not been identified) have had a mild infection with no fatalities [60]. Indeed, this scenario is more compatible with *R. parkeri* rickettsiosis, which has epidemiological support for our serological results.

Although the Goiás State has not been considered endemic for *R. rickettsii* and *R. parkeri* like other Brazilian regions [18,61], a combination of ecological and anthropogenic factors may create favorable conditions for the circulation of such agents, particularly due to the overlapping distribution of competent vectors, wild boars and capybaras, the amplifying host of *R. rickettsii* in the Cerrado biome [18,58,62,63]. Consequently, people engaged in the control of wild boars are at direct risk of contact with the BSF vector, given that this activity entails direct contact with the wild boar, especially following culling [23].

## 5. Conclusions

The present study has indicated that the Eurasian wild boar has been exposed to some species of *Rickettsia* in the Cerrado biome in Goiás State, midwestern Brazil. Although most of the seroreactive wild boars presented low endpoint titers with cross-reactions to two or more *Rickettsia* species, a few animals presented serological evidence of possible exposure to *R. parkeri*, *R. amblyommatis,* and *R. bellii*. There was no serological evidence of wild boar exposure to the BSF agent *R. rickettsii*, which is transmitted to humans mainly by *A. sculptum*. Given that *A. sculptum* is the most common tick species infesting wild boars in the Cerrado biome, the serological results of the present study indicate that the southern region of Goiás is not endemic for BSF if one considers wild boars as suitable sentinels for BSF. Further studies should be conducted to establish whether wild boars are competent reservoirs and amplifiers of both *Rickettsia* spp. or may be used only as sentinels for rickettsial infection surveillance.

## Figures and Tables

**Figure 1 animals-14-02224-f001:**
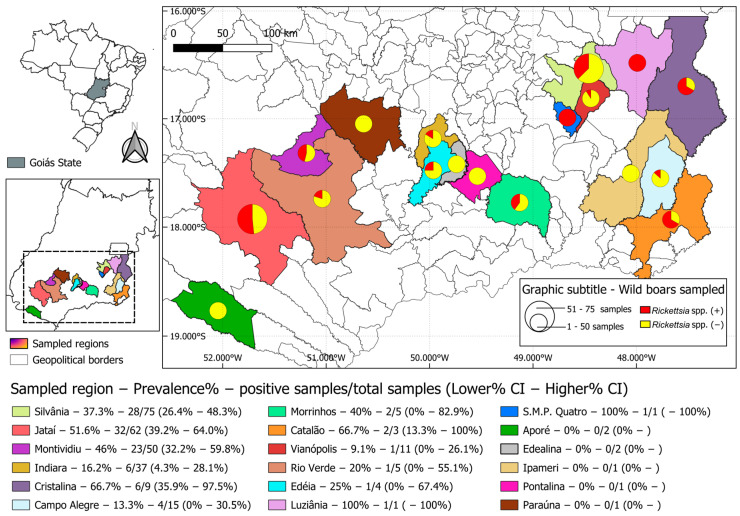
Seroprevalence (95% CI) results of anti-*Rickettsia* spp. antibodies in wild boars in the Cerrado biome from Goiás state, midwestern Brazil.

**Table 1 animals-14-02224-t001:** Seroreactivity for four species of *Rickettsia* in 285 wild boars sampled in the Cerrado biome of the Goiás State, midwestern Brazil.

Municipalities	No. Tested Samples	No. of Wild Boars Reactive to Each *Rickettsia* Species (Range of Endpoint Titers)				No. of Animals with PAIHR *
		*R. rickettsii*	*R. parkeri*	*R. bellii*	*R. amblyommatis*	
Silvânia	75	6 (64–256)	17 (64–256)	2 (64–128)	16 (64–512)	1 *R. bellii*; 5 *R. amblyommatis*
Jataí	62	10 (64–256)	30 (64–256)	4 (64–256)	14 (64–256)	7 *R. parkeri*; 3 *R. amblyommatis*
Montividiu	50	6 (64–256)	14 (64–256)	14 (64–256)	3 (128)	6 *R. parkeri*; 4 *R. bellii*
Indiara	37	3 (64–128)	5 (64–128)	1 (64)	2 (128–256)	-
Cristalina	9	2 (64–128)	6 (64–256)	-	-	1 *R. parkeri*
Campo Alegre	15	-	2 (64–128)	-	-	1 *R. parkeri*
Morrinhos	5	-	1 (128)	-	1 (64)	1 *R. parkeri*
Catalão	3	-	-	2 (128–256)	-	2 *R. bellii*
Vianópolis	11	-	1 (128)	-	-	1 *R. parkeri*
Rio Verde	5	-	-	1 (128)	-	1 *R. bellii*
Edéia	4	-	-	1 (256)	-	1 *R. bellii*
Luziânia	1	-	-	1 (64)	-	-
^•^ S. M. P. Quatro	1	-	-	1 (64)	-	-
Aporé	2	-	-	-	-	-
Edealina	2	-	-	-	-	-
Ipameri	1	-	-	-	-	-
Pontalina	1	-	-	-	-	-
Paraúna	1	-	-	-	-	-
TOTAL ^#^	106/285 (37.2%)	27/285 (9.5%)	76/285 (26.7%)	27/285 (9.5%)	36/285 (12.6%)	9 *R. bellii*; 8 *R. amblyommatis*; 17 *R. parkeri*

* PAIHR: Possible antigen involved in homologous reaction. A PAIHR was determined when an endpoint titer to a *Rickettsia* species was at least 4-fold higher than those observed for the other *Rickettsia* species. -: no sample was reactive at the 1:64 serum dilution. ^#^: No. of seroreactive animals/No. of tested animals (% seroreactivity). ^•^ São Miguel do Passa Quatro.

## Data Availability

The map produced for this work was generated from information obtained on the website of the Brazilian Institute of Geography and Statistics (IBGE) and the data can be found at https://www.ibge.gov.br/geociencias/informacoes-ambientais/estudos-ambientais/24252-macrocharacterizacao-dos-recursos-naturais-do-brasil.html, accessed on 30 June 2024.

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
