# Peer review of "Serological and Molecular Survey of Rickettsial Agents in Wild Boars (Sus scrofa) from Midwestern Brazil"

_animals, 2024, doi:10.3390/ani14152224_

Round 1

Reviewer 1 Report

Comments and Suggestions for Authors

Dear authors, congratulations on your highly interesting manuscript. This thorough study on the seroprevalence of Rickettsia spp. in a large sample of Sus scrofa in the Cerrado biome further increases current knowledge on the role of wild boars as sentinels for spotted fever illness. I would like to point out some specific concerns, as follows.

Title, Simple Summary, Abstract

Line 2 (Title) – Please add the scientific name of the species between parentheses.

Lines 37/38 – remove the sentence beginning at “The largest number (…)”, and merge it into the previous sentence, for example “(…) antigens tested, the largest number in this type of study”.

Keywords

Do not use IFA as a keyword, I would suggest replacing it with “immunofluorescence assay”.

Introduction

            Line 52 – To this reviewer’s best knowledge, wild boars currently live on every continent except Antarctica but are only native to Asia. As such, I would remove “(…) in their natural environment such as Europe (…)”  

            Line 64 – Remove the repeated word “collected”

Materials and Methods

Line 75-77 – No references are mentioned concerning these epidemiological statements, please add them.

Line 82 – Add detailed information concerning blood centrifugation

Line 92 – I would suggest changing the expression “as described elsewhere” to “as described in the literature”, or similar.

Results

            Line 112 – Remove this information as this is already explained, as it should, in the Material and Methods section.

            Lines 132/133 – This information should be given in the Material and Methods section, not in the current section.

Discussion

Line 143 – Choose to mention “wild boars” or “S. scrofa”, not both.

Line 160 – I would suggest changing the expression “our findings” to “the current findings”, or similar

Line 167 – I would suggest changing the expression “we found” to “it was found”, or similar

Table 1 – This table should be placed before the Discussion section.

Comments on the Quality of English Language

Some minor english editing is needed, for example:

- placing/lack of commas;

- singular/plural (e.g. wild boars instead of wild boar on several occasions; line 213 "point out" instead of "points out");

- "Because previous studies report" instead of "Because there are previous studies reporting";

- "Brazilian" instead of "brazilian" on several occasions;

- Add "the" before "Brazilian government";

- Remove expressions such as "In fact".

Author Response

Dear Reviewer 1,

Best regards, 

Reviewer 2 Report

Comments and Suggestions for Authors

This study aimed to determine the seroprevalence of Rickettsia spp. in a large sample of wild boars and to test ticks collected from these animals for the presence of rickettsial DNA in the Cerrado biome of midwestern Brazil. The main contributions include the detection of anti-Rickettsia spp. antibodies in 37.2% of wild boar serum samples, indicating significant exposure to rickettsial agents, the identification of Amblyomma sculptum ticks, all of which tested negative for rickettsial DNA, and the identification of the highest specific seroreactivity for R. parkeri, a known SFG agent in Brazil. The study's strengths lie in its large sample size, the largest of its kind in South America, and its comprehensive approach to assessing the interaction between wild boars and Rickettsia spp. within the region, analysing both the Wild boars and their ticks.

The study is overall well-conducted and well-written, accompanied by a high-quality image and table that summarize the information effectively.

Despite some minor English improvements needed, my main suggestion for the study is to elaborate on certain parts of the introduction and discussion, which seem a little short and incomplete.

In the introduction I suggest to further explaining the spotted fever illness caused by Rickettsia rickettsii and by Rickettsia parkeri in Brazil: their symptoms, mortality rates, known distribution, known reservoir species and if Wild boars have any reported role in spreading the disease. I would also suggest adding some infectious agents that Wild boars are known to be important reservoirs, like the ones described in the study: doi: 10.3390/ani13111743.

In the discussion I suggest to explain or theorize why the prevalence of seroreactive wild boars to Rickettsia spp. in your study (37.2%) is much lower than in previous studies (77.4% and 72.5%). Could this discrepancy be explained by the fact that the other studies also used a fifth antigen (for Rickettsia rhipicephali), geographical differences in the sampling process, or the significantly larger sample size in your study?

In the line 149, specify where the sampling for the study of Kmetiuk et al. (2019) took place, like you did to the study of Sousa et al. (2024).

In the line 157 better clarify which Rickettsia agent those prevalences refer to.

Emphasise more clearly that the highest specific seroreactivity in the previous studies of Wild boars (Kmetiuk et al. 2019 and Sousa et al.2024) was for R. rickettsii, the agent of the Brazilian spotted fever (BSF), additionally in your study, the highest specific seroreactivity was for R. parkeri, other known agent of spotted fever illness in humans in Brazil. Further discussing the importance of that discoveries and role that Wild boars could have in the surveillance and dissemination of that diseases. Include the improved discussion on this topic in the conclusion of your work.

Comments on the Quality of English Language

Some English improvements are needed. For example:

In the line 57, “have had” should be replaced by “have been”.

The sentence of the lines 80-82, should be revised to “Hunting of wild boars has been permitted in Brazil, where they are considered invasive exotic fauna. In Goiás State, this activity is coordinated by the Agricultural Defense Agency (Agrodefesa)”

Author Response

Dear Reviewer 2,

Best regards, 

Reviewer 3 Report

Comments and Suggestions for Authors

Estimated Authors,

thank you for the work done and to contribute to Animals with your study.

The manuscript of Jalowitzki de Lima et al., investigates the exposure of wild boars to rickettsial agents across different areas of Goiás State. The results showed a significant seroprevalence of rickettsial exposure involving different wild boar populations. Seroreactivity was mainly associated to the causative agents R. parkeri, R. amblyommatis or R. bellii. A limited sample of on-host adult ticks were also collected, identified as A. sculptum and screened for rickettsial-DNA detection, obtaining negative results for all the specimens analysed.

In general, the manuscript is well structured and comprehensively written. The used methodology is adequate for the purpose of the study and correctly described, as well as the results section. However, the introduction and the discussion are the weakest sections, in terms of details. I appreciate the Authors to go to the point, directly; however, I would have liked a more detailed description of some aspects. For example, in the Introduction section, it would be helpful to give details on the disease in humans, by specifying the incidence and the geographic distribution of the disease in humans in Brazil or at least in the study State. Also, it would be interesting to describe the main tick vectors that play a role as reservoir or carriers of rickettsial agents in the study area, and which of these are commonly involved in tick bites in humans. All these details will let to clearly comprehend the epidemiology of rickettsioses in your context.  With regard to the Discussion section, please integrate as requested in the comments below.

Minor comments

L155-58: Could you give some explanation of these differences compared with your findings? It is interesting that you found a significant lower prevalence in the occurrence of antibodies against R. rickettsii, despite the larger sample size you analysed. Differences in the type of habitats surveyed could be a possible explanation? This ecological aspect has a direct impact in the distribution of the tick vectors? Could be hypothesize the role of tick vectors, other than A. sculptum, involved in the cycle transmission of this rickettsial agent? Regarding the higher seroprevalence found for R. parkeri, could you hypothesize that wild boar could be commonly exposed to Amblyoma tick bites other than A. sculptum-tick bites? Also here, you could highlight differences in the habitat type surveyed between the study sites. In fact, almost all the regions, where the positivity to R. parkeri occurred, are characterized by the same habitat type (Figure 1). It would be interesting to include some reasoning about this ecological aspect within the discussion.

L174-77: Could you be more specific? Which further investigations? About what?

L198: Which Amblyoma species apart from A. sculptum? Please, specify it.  You did not mention any species in the entire paragraph. I think that this information is relevant to reinforce the potential of this rickettsial species to cause disease in humans (in the case that the tick vectors would be commonly involved in tick bites, and even though the two agents are not considered pathogenic, yet). Also include some hypothesis about the difference in occurrence of these two rickettsial agents, maybe related to habitat type, different tick species involved in their maintenance, etc.

L211-12: You cannot state this from your results. You only collected ticks from a limited number of wild boar, concentrating in three specific areas. So you results are not representative in this aspect. Modify the sentence accordingly.

L216-18: I do not understand the relation between the titers in wild boar and human cases. Are you sure that it is not endemic? You find it at a prevalence of almost 10%, that means 1 out of 10 wild boar are exposed in your area. How many ticks can wild boar harbour? How many ticks are infected by R. rickettsii? I think that further assessments on ticks and wild boar should be carried out to evaluate the endemicity of R. rickettsii in your area.  Human cases could be underestimated?

Author Response

Dear Reviewer 3,

Best regards, 

Reviewer 4 Report

Comments and Suggestions for Authors

Dear authors, follow my comments and suggestions:

- The introduction is shallow in terms of characterizing the problem, not providing sufficient theoretical foundation on the situation of Sus scrofa in Brazil and studies that mention its importance as a reservoir of pathogens in Brazil. The context of Sus scrofa and some of its interactions with parasites in Brazil is fundamental to understanding the authors' justification for carrying out such a study.

- Line 75-77: Unnecessary information, as it is not used in the study.

-  Is not clear from the study why ticks were only collected from the year 2023. Was it a design error?

- Why didn't you test the positive IFA samples by PCRfor Rickettsia too?

- Line 112: This was already clear in the methods.

- The text in item 3.1 needs writing refinement. The data appears scrambled.

- There are citations outside the magazine's standard in the discussion.

- The results are poor in evidence, especially due to the lack of molecular tests in seroactive animals. The discussion does not address central problems, such as the risk of human exposure, even through hunting these animals. At least one paragraph should discuss and address more emphatically that the interaction between humans and wild boars favors the transmission of ticks and, consequently, pathogens associated with them. The study is relevant, but it appears that the authors only did parts of the study or did not design it well.

Author Response

Dear Reviewer 4,

Best regards, 

Round 2

Reviewer 3 Report

Comments and Suggestions for Authors

Dear Authors,

thank you for implementing all the recommendations I provided during the previous round of the peer-review process. I am so grateful for your efforts in this regard.

On this occasion, I have no concerns regarding the content, which I believe to have been addressed in sufficient depth. However, some formal changes to the terminology and the language employed should be advisable to enhance clarity and comprehension.

Please, find below a list of minor comments, which includes my proposed options highlighted in red. In general, it would be advisable to revise the English language of the entire manuscript.

With kind regards,

Reviewer #3

Introduction

L65-67: At the beginning of the sentence you refer to diseases, then you nominate pathogens… Please be consistent.

L86-90: This paragraph could be joined with the next paragraph as both address the role of wildlife in the disease cycle. Also, some rephrasing is required. See my suggestion: “In the natural environment, the capybara (Hydrochoerus hydrochaeris) is regarded as the primary amplifying host for the spotted fever agent, R. rickettsii. Conversely, the rice rat (Euryoryzomys russatus) has been identified as a competent amplifying host for R. parkeri, particularly for immature stages of the tick vector, Amblyomma ovale.”

Materials and methods

L115-132: I am unsure ‘slaughtered’ would be the correct term to used here, if the animals were hunted the correct term to use may be ‘culled’. Please, revise this throughout the manuscript. In view of new changes provided in the current version of the manuscript. I propose to use some formal changes in section 2.2. Sampling to improve clarity and comprehension:

“Hunting of wild boars has been permitted in Brazil as an invasive exotic fauna, with the coordination of this activity falling under the responsibility of the Agricultural Defense Agency (Agrodefesa) in the Goias State. Overall, 285 blood samples were collected from culled free-ranging wild boar in different locations within Goias State in collaboration with exotic fauna controllers (i.e., hunters) between 2018 and 2023. Following the placement of tubes in appropriate packaging, the samples were transported to the local Agricultural Defense Agency and immediately shipped to the University Laboratory. Serum samples were aliquoted and stored at -20ºC until they could be processed for serological analysis. In addition, a convenience sample of 45 ticks was collected from four free-ranging wild boar that were culled in the municipalities of Silvânia, Edealina and São Miguel do Passa Quatro. This was conducted over the course of eight on-field incursions, which lasted approximately 67 hours, during 2023. The rationale behind the decision to sample only four animals for tick collection was based on the inherent unpredictability and limited productivity of on-field sampling of free-ranging wild boar.”

In L121-122 You had added: “As limitations, no control was made during wild boar hunting, samplings, serum and clot separation, sample identification and shipment.” I do not understand what you mean with this sentence, no control of what?

Moreover, in L123-124 you added: “Despite a high frequency of seropositive animals, results herein made be higher than observed.” I do not know why to specify this here… You are describing your methodology; this sounds part of the discussion or conclusion… Please move it from here.

L151: “description and taxonomic criteria”

Discussion

L214-218:These results, coupled with the current findings of 17 animals with seroreactivity that may be homologous to R. parkeri, suggest that wild boars have been exposed to R. parkeri in Goiás state. Additionally, in Minas Gerais, another state within the Cerrado biome, and in the Atlantic Forest biome, there is a heightened exposure to R. rickettsii among wild boars [22,23].”

L219-226: The first sentence of the paragraph is quite long, preventing for the clarity of the idea behind. Please, consider the following changes: “Discrepancies in seropositivity rates for Rickettsia spp between the present study and previous findings [22, 23] can be attributed to several factors, including differences in sample size, wild boar populations, and geographic and biome locations. Such differences may significantly impact regional variations of tick species and amplifier hosts, leading to disparate exposure to rickettsial agents. Edaphoclimatic conditions may also influence the presence of tick hosts and, consequently, the rickettsial cycle [18,39]. Therefore, future studies should address the R. parkeri ecology and ecoepidemiology, assessing the dynamics between wild boar and rickettsial agents.”

L260-263: “Serological results presented here for R. amblyommatis may be explained by its common detection in ticks. This rickettsial agent has already been described in 34 tick species worldwide [56], including A. sculptum, Amblyomma oblongoguttatum and Amblyomma scalpturatum, which have been identified as parasitising pigs in Brazil”.

L265-270: “The identification of all ticks as A. sculptum was expected, given the narrow host specificity of this tick vector, which has been previously detected in wild boars in Brazil [2,22,24], and the regions from which the specimens were collected (Cerrado biome) [25, 59]. Notwithstanding the absence of Rickettsia spp. DNA detection in the collected ticks, the potential risk cannot be ruled out, particularly in light of the limited sample size and the observation that less than 1% of ticks in nature were infected by R. rickettsii [37].”

L274-282: “No association was possible to establish between the serological findings pertaining to R. parkeri and the tick species identified. However, ticks that act as vectors of this agent have been documented in wild boars and domestic pigs (S. scrofa). Amblyomma triste has been described parasitizing wild boars in Argentina, suggesting that these animals may contribute to the maintenance of this tick species in that environment [48]. A. ovale has been documented parasitizing wild boars in transition areas between the Cerrado and Atlantic Forest biomes, where hemopathogens of the Anaplasmataceae family have been detected [24]. Furthermore, parasitism of A. ovale has been reported in domestic pigs from areas of the Amazon Rainforest [57].”

L283-285: The meaning of the sentence is confusing. Please, could you rearrange it?

L291-294: “Seropositivity for R. bellii may be associated with the prior detection of the pathogen in A. dubitatum, in addition to the high serological titers in dogs and capybaras within the Goiás State [51]. A. dubitatum has been observed parasitizing wild and domestic pigs [48], indicating the possibility of exposure to R. bellii.”

L295-297: “Our findings demonstrate that only 9.5% (27/285) of the wild boars were seroreactive to R. rickettsii, 295 with endpoint titers ranging 64 – 256. This evidence suggests that the southern region of Goiás state may not be endemic to BSF.”

L302-308: “…a combination of ecological and anthropogenic factors may create favourable conditions for the circulation of such agents, particularly due to the overlapped distribution of competent vectors and capybaras, the amplifying host of R. rickettsii in the area [18,58,62,63]. Consequently, people engaged in the control of wild boars are at direct risk of contact with the BSF vector, given that this activity entails direct contact with wild boar, especially following culling [23]”

Comments on the Quality of English Language

Formal changes to the terminology and the language employed should be advisable to enhance clarity and comprehension. In general, it would be advisable to revise the English language of the entire manuscript.

Author Response

Dear Reviewer 3, 

Best regards, 

Reviewer 4 Report

Comments and Suggestions for Authors

The authors' changes and explanations were enough for me.

Author Response

Dear Reviewer 4, 

Best regards, 
